# The Effects of Stimulus Repetition Rate on Electrically Evoked Auditory Brainstem Potentials in Postlingually Deafened Adult Cochlear Implant Recipients

**DOI:** 10.3390/jcm12227188

**Published:** 2023-11-20

**Authors:** Oliver C. Dziemba, Tina Brzoska, Thomas Hocke, Friedrich Ihler

**Affiliations:** 1Departement of Otorhinolarygology, Head and Neck Surgery, University Medicine Greifswald, 17475 Greifswald, Germany; tina.brzoska@med.uni-greifswald.de (T.B.); friedrich.ihler@med.uni-greifswald.de (F.I.); 2Cochlear Deutschland GmbH & Co. KG, 30539 Hannover, Germany; thocke@cochlear.com

**Keywords:** objective measurement, cochlear implant, differential diagnostics

## Abstract

**Background:** By using outcome prediction scores, it is possible to distinguish between good and poor performers with cochlear implants (CI) after CI implantation. The reasons for poor performance, despite good basic conditions, can be manifold. On the one hand, the postoperative fitting may be inadequate; on the other, neurophysiological disease processes may impair speech understanding with a CI. These disease processes are not yet fully understood. In acoustics, it is known that the auditory brainstem responses (ABR) and their latencies and amplitudes allow differential diagnosis based on reference values for normal-hearing individuals. The aim of this study was to provide reference values for electrically evoked brainstem responses (EABRs) in terms of rate-dependent latencies and amplitudes. **Methods:** 20 ears of 18 experienced adult CI recipients with a predicted and measured good postoperative word recognition score were recruited from the clinic’s patient pool. In the same stimulation mode and intensity we measured latencies and interpeak-latencies of EABRs and electrically evoked compound action potentials (ECAPs). With a defined supra-threshold stimulation intensity above the individual ECAP threshold, we applied stimulation at several rates between 11 and 91 stimuli per second. **Results:** We found rate dependences for EABR latency t3 and t5 in the order of 0.19 ms and 0.37 ms, respectively, while ECAP was not affected by rate. Correspondingly, the interpeak intervals’ rate dependences for t5−t1, t5−t3 and t3−t1 were of the order of 0.37 ms, 0.18 ms and 0.19 ms. Comparing the EABR amplitudes between the stimulation rates 11/s and 81/s, we found that at 81/s the amplitudes were significantly reduced down: to 73% for A3 and 81% for A5. These rate dependences of latency and amplitude in EABR have characteristics comparable to those of acoustic ABR. **Conclusions:** These data may serve to provide reference values for EABR and ECAP latencies, interpeak intervals and amplitudes with respect to stimulation rate. Altered response patterns of ECAPs and EABRs to normalised stimulation modes could be used in the future to describe and classify neuropathological processes in a better-differentiated way.

## 1. Introduction

Cochlear implantation is an established therapy for sensorineural hearing loss if hearing aids and other solutions fail to restore speech recognition [1,2]. Recent studies reported on successful cochlear implant (CI) provision for patients with hearing losses from 50 to 80 dB [3,4,5,6]. However, even in these patients, with good preconditions for postoperative word recognition—and even more in the established patient population with no preoperative word recognition [6]—some challenges still remain. Recent studies and opinions [3,7,8,9,10] indicate a lack of audiological differential diagnosis in these patients and highlight the observation that “the broad array of factors that contribute to speech recognition performance in adult CI users suggests the potential both for novel diagnostic assessment batteries to explain poor performance, and also new rehabilitation strategies for patients who exhibit poor outcomes” [7].

To our knowledge, there is no generally agreed classification of CI recipients with respect to performance or to speech perception in general. A prediction model recently introduced by Hoppe et al. [3] for the expected postoperative word recognition score at conversation level, WRS_65_(CI), after six months of CI use would allow such a classification. Thus, failure to reach this goal can easily be assessed by routine clinical audiometry [3,6]. “Unexplained poor performance” may be defined as applying to CI recipients whose WRS_65_(CI) does not meet the predicted score according to this model. Such cases can be observed with an incidence of around five percent in a population with residual preoperative word recognition score (WRS) [6] on the basis of a 20-percentage-points (pp) difference (WRS_GAP_) between prediction and measurement in monosyllable test scores. Users who reach the predicted score later than six months after implantation (e.g., twelve months later) would not be covered by this definition [6]. More recently, in a study by Dziemba et al. [11] such a definition was applied in order to identify systematic differences in postoperative fitting of CI systems in a group of well and poorly performing CI recipients, namely the differences in audibility and the loudness growth function measured by categorical loudness scaling. An additional application of the prediction model [3] could be the interpretation of electrophysiological measurements based on prior classification of groups of recipients in respect of the WRS_GAP_. Other recent work [8,12,13,14,15] led to the proposal and use of a setting for electrically evoked auditory brainstem responses (EABR) mimicking the established acoustic broadband click. Reference values for latencies were assessed by including only CI recipients with word recognition score with cochlear implant at 65 dB (WRS_65_(CI)) of 50% or more [13]. This approach led to improved differential diagnosis for CI recipients and improved intraoperative assessment by using objective methods like electrically evoked compound action potentials (ECAP) and EABR [13,15,16].

Some characteristics of CI recipients, such as rate dependence of electrophysiological measurements, indicate a potential for improvement in differential diagnostics. In the acoustic modality, rate effects in auditory brainstem responses (ABR) are already well described [17,18]. Jiang et al. [18] reported age-dependent latencies and interpeak intervals in children as consequences of developmental effects. In our opinion, this measure of auditory synaptic efficacy [18] can be transferred to differential diagnostics in CI recipients to provide further explanation of unexpectedly poor WRS_65_(CI) values. We expect that certain damage mechanisms in hearing-impaired subjects and CI recipients may have a similar effect on rate dependence of EABRs.

Consequently, the goal of this study was to provide reference values for rate-dependent EABR in CI recipients. By including only CI users who met the predicted values of WRS_65_(CI), we aimed to open a window for differential diagnostics in CI recipients with unexpected and unexplained poor postoperative WRS_65_(CI).

## 2. Materials and Methods

### 2.1. Research Subjects

This prospective investigation included five subjects in the pilot phase, and thereafter a further 20 subjects, according to a power calculation based on the results of the pilot phase. The power calculation was based on a effect size of 0.45 ms, which was the mean of the rate dependence of latency t5 of the pilot measurements (standard deviation of 0.22). We used a balanced one-way analysis of variance power calculation. The significance level was set to 0.05 and the power was set to 0.95.

The study was approved by the Ethics Committee at the University Medicine Greifswald on 10 August 2021 (BB 120/21), and all procedures were in accordance with the ethical standards of the institutional and national research committee and with the 1964 Helsinki declaration and its later amendments or comparable ethical standards. The study was registered in the German Clinical Trials Register (DRKS00026195 https://drks.de/search/de/trial/DRKS00026195 (accessed on 16 October 2023)).

Participants were recruited from the clinic’s patient population. Inclusion criteria were:Adulthood (minimum age 18 years) at implantation;Implant type: CI24RE, CI400 series, CI500 series, or CI600 series (Cochlear™ Limited, Sydney, Australia);Implant in specification according to European Consensus Statement on Cochlear Implant Failures and Explantations [19];WRS_65_(CI) in the upper three quartiles according to classification of Hoppe et al. [20];Willingness and ability to participate in the study.

Exclusion criteria were:Known mental handicap;Known central hearing disorders;Short cut or open circuit of intracochlear electrodes 11 and/or 18.

Demographic information for these patients is provided in Table 1. Bilateral implantation was not an exclusion criterion; in those two cases, both ears were included in the analysis separately (#098/#107 and #271/#275). The mean age at the time of inclusion in the study of the participants was 59 years (minimum age 38 years, maximum age 74 years). The participants had a mean hearing experience, usage of CI, of 51.4 months (min. = 1 month, max. = 146 months).

### 2.2. Electrophysiological Measurements

To measure rate dependences of latencies and inter-peak latencies of EABRs, a quasi-simultaneous measurement of ECAPs and EABRs is needed. Here, it is essential to use the same stimulation mode and the same stimulation intensity for both assessments, to ensure compatibility of the data. Therefore, Dziemba et al. [12] introduced an intracochlear stimulation mode for the Nucleus^®^ CI system (EABR_CI_Stim). They used electrode 11 as a stimulation-active and electrode 18 as a stimulation-indifferent electrode, with a pulse width of 100 µs. This EABR_CI_Stim facilitates an electrical excitation covering a median length of about 80% of the length of the implanted CI electrode array. Since ECAP and EABR are recorded with opposite polarity, the inter-peak latencies were determined between the negative peak of the ECAP and the corresponding positive peaks in the EABR, measured at the same stimulation intensity.

In order to avoid possible intensity-dependent effects, a defined supra-threshold stimulation intensity of 20 current levels (CLs) above the individual ECAP threshold, measured with the EABR_CI_Stim, was set. The measurements in all subjects followed the same procedure, as described below.

#### 2.2.1. ECAP Measurements

For the unconditional avoidance of uncomfortable loud stimulation the loudest acceptable presentation level (LAPL) using the EABR_CI_Stim was estimated subjectively in a first step.

The second step was the identification of the most appropriate recording-active electrode according to Dziemba et al. [12]. Therefore, ECAP was measured at LAPL by stimulating all intra-cochlear electrodes, except electrodes 11 and 18, sequentially by using the extracochlear electrode (plate) MP2 as recording-indifferent electrode. The electrode with the largest ECAP amplitude at LAPL was selected as the best recording-active electrode.

In the third step, an ECAP amplitude-growth function was measured up to LAPL with the values found previously. The visual ECAP threshold was read out, taking into account a minimum signal-to-noise ratio for ECAP measurements according to Hey et al. [21].

Finally, the rate-dependent ECAPs were measured by using a stimulation intensity of 20 CL above the previously found threshold at stimulation rates of 11, 41, 81 and 91 stimuli per second.

#### 2.2.2. EABR Measurements

All EABR measurement series were performed in the same stimulation mode as for the rate-dependent ECAP, using the EABR_CI_Stim described above. The Eclipse system (Interacoustics, Middelfart, Denmark) was used to record the rate-dependent EABRs. Synchronisation between the CI system and the EABR device was achieved through a TTL-compatible trigger signal. This was sent via a commercially available cable (3.5 mm jack) from the programming interface of the CI system to the EABR recording system. The marking of all the measured potentials (ECAP and EABR) was performed according to Atcherson and Stoody [22]. To avoid ambivalence in picking peaks, they recommended that the rightmost sample be used for marking the positive peaks and the leftmost sample be used for marking the negative peaks. The labelling and numbering of the marked potentials was performed according to Jewett and Williston [23].

### 2.3. Statistical Analysis

We used boxplots, as defined by Tukey [24], for the graphical representation of the measured values.

For each user, the set of measurement data are a connected, non-normally distributed sample. Furthermore, there is no variance homogeneity of the data. Therefore, a non-parametric test must be used; we chose the Friedman rank sum test as being the most appropriate. As a post hocanalysis, we used the test of multiple comparison after Friedman test.

All statistical tests and figures were conducted with R [25] and RStudio [26].

## 3. Results

### 3.1. Latencies

The latencies t1, t3 and t5 of rate-dependent ECAP and EABR are shown in Figure 1. While no rate effect on latency t1 was seen for the ECAP (p=0.07), we found significant mean rate effects for latency t3 (0.19 ms) and t5 (0.37 ms) The post hoc analyses of the rate effects of t3 and t5 are summarised in Table 2 and Table 3.

### 3.2. Interpeak Intervals

The interpeak intervals t5−t1, t5−t3 and t3−t1 of rate-dependent ECAPs and EABRs are shown in Figure 2. We found significant rate effects for all the interpeak intervals analysed. The interpeak interval t5−t1 shows a rate effect of 0.37 ms, while the interpeak interval t5−t3 is shows a rate effect of 0.18 ms. The mean rate effect on interpeak interval t3−t1 is 0.19 ms. The analyses are summarised in Table 4, Table 5 and Table 6.

### 3.3. Amplitudes

The amplitudes A1, A3 and A5 of rate-dependent ECAPs and EABRs are shown in Figure 3. While for ECAP, there is no rate effect on A1 (p=0.26), we found significant detrimental rate effects, for A3 and A5, with respective mean reductions of 73% and 81%. The post hoc analyses of the rate effects of A3 and A5 are summarised in Table 7 and Table 8.

## 4. Discussion

In accordance with the study’s goals, we investigated the rate dependences in our population of CI recipients, all of whom had monosyllabic word recognition within the upper three quartiles according to the classification put forward by Hoppe et al. [20]. We found rate dependences for EABR latency t3 and t5 in the order of 0.19 ms and 0.37 ms, respectively, while ECAPs were not affected by rate. Correspondingly, the interpeak intervals’ rate dependences for t5−t1, t5−t3 and t3−t1 were found to be in the order of 0.37 ms, 0.18 ms and 0.19 ms. Jiang et al. [18] described the change in rate dependence in acoustic ABR as an effect of the maturing auditory pathway in children of various ages. In adults, the latency changes with rate are probably related to synaptic adaptation [17]. With respect to the amplitudes, Campbell et al. [27] have stated that the change in wave V of acoustic ABR does not decrease at 81/s by more than 28% compared with the amplitude at 11/s. However, in our population, we found significant detrimental rate effects: a reduction down to 73% for A3 and down to 81% for A5. This is within the range for rate-dependent changes found for wave V in acoustic ABR [27]. To summarise, these reference values for EABR and ECAP latencies, interpeak intervals and amplitudes provide a basis for possible differential diagnoses after cochlear implantation.

We hypothesize that in postlingually deafened adults with CI, larger changes in amplitudes and latencies due to rate (in comparison with references values) can be interpreted as pathological effects. The values shown above can be regarded as reference values. Pathologies may then be revealed in significant deviations from them. For example, in patients with auditory neuropathy spectrum disorder the dyssynchronous neural activity may affect temporal encoding of electrical stimulation from a cochlear implant [28]. Even though Fulmer et al. [28] investigated the recovery function of ECAP, one may reasonably assume that EABR measurements and their rate dependences will be affected by these pathological mechanisms as well. Continuing this line of thought, we would argue that, compared with ECAP, EABR assesses the higher levels of the auditory pathway as well, and therefore appears to offer a valuable complement within differential diagnostics. However, while ECAP can be considered to provide tonotopic information, the EABR as applied in this study will provide integrated information about the status of the auditory pathway. This differential diagnostic pattern is especially important for the most recent CI population [3,4,5,6] with higher preoperative speech recognition scores. In this patient population, a highly predictive outcome was observed [3,6] compared with the established patient population with no preoperative speech perception [6,29]

Consequently, if in the patient population with good audiometric preoperative conditions [6] the prediction cannot be achieved, an underlying pathology of the auditory pathway may be suspected. Moreover, approaches utilising advanced measurements of ECAPs [8,30,31,32,33], and the assessment of EABR and its rate dependences might be suitable in analogy to the findings in acoustic ABR. With values up to 0.28 ms the standard deviation for the interpeak interval t5−t1 seems to be slightly higher than the 0.23 ms found by Campbell et al. [27]. A more thorough analysis will be needed in future studies.

Assuming a higher standard deviation (which still has to be confirmed), this may have its root cause in the inclusion criteria of the CI population. For acoustic stimulation, normative values, and the population in which to assess them, are easy to define as one has by definition to include normal-hearing subjects. In the case of CI recipients, the definition of a reference group is far more challenging. There are no generally agreed criteria for the derivation of a reference group. Consequently, the reference values provided by this study can potentially be improved by better outcome prediction models and, based on this, a narrower patient selection.

Recently, Hoppe et al. [6] applied the criterion “unexpectedly poor speech perception”, defined as monosyllabic speech recognition ≥ 20 pp lower than predicted after six months, in order to discuss the time course of such cases. The six months were derived from study which found that 90% of the final score is achieved after 6.9 months. Even if in that study [6] the majority of subjects who were poor perfomers after six months nonetheless reached the target value after a longer time period, there remain 5% of cases in which the prediction is not reached in the long run. The aim of differential diagnostics using EABR would be to differentiate between a patient’s intrinsic root causes for unexpectedly poor speech perception (pathologies) or causes in which the fitting of CI system also plays a part [11]. Future studies with a focus on the time course of postoperative speech recognition with respect to different pathologies (once these are confirmed) will be needed to refine the diagnostics using EABR.

## 5. Conclusions

The rate-dependences of latency and amplitude in EABR have characteristics comparable to those of acoustic ABR. Consequently, EABR may potentially support differential diagnosis in CI recipients with an outcome below expectation. The results of this study may serve to provide reference values. Pathological issues of the peripheral auditory pathway hindering a postoperative increase in speech perception and CI outcome in general can be excluded or confirmed.

## Figures and Tables

**Figure 1 jcm-12-07188-f001:**
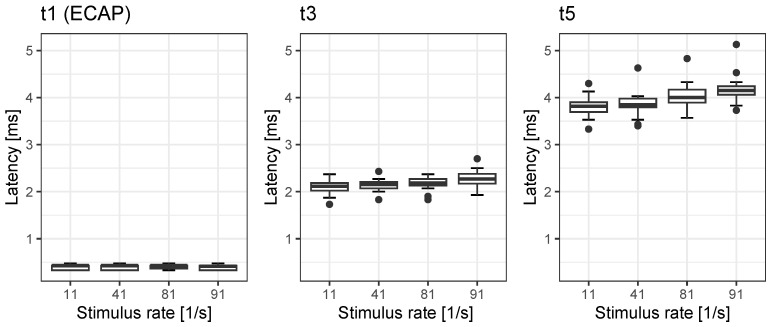
Latencies of all measured potentials. The boxes show medians and quartiles; the whiskers show the maximum value within 1.5 IQR (1.5 times the interquartile range). Filled circles show the outliers.

**Figure 2 jcm-12-07188-f002:**
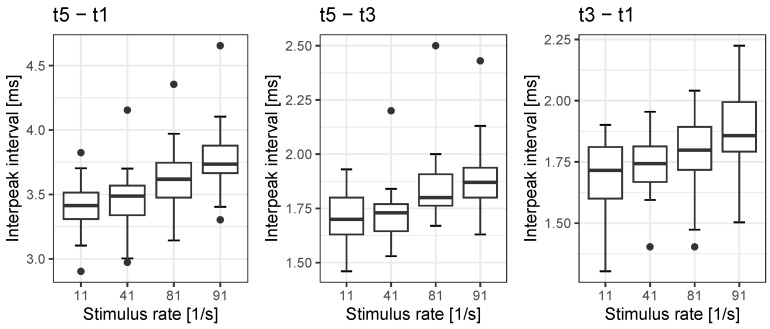
Interpeak intervals of all measured potentials. The boxes show medians and quartiles; the whiskers show the maximum value within 1.5 IQR. Filled circles show the outliers.

**Figure 3 jcm-12-07188-f003:**
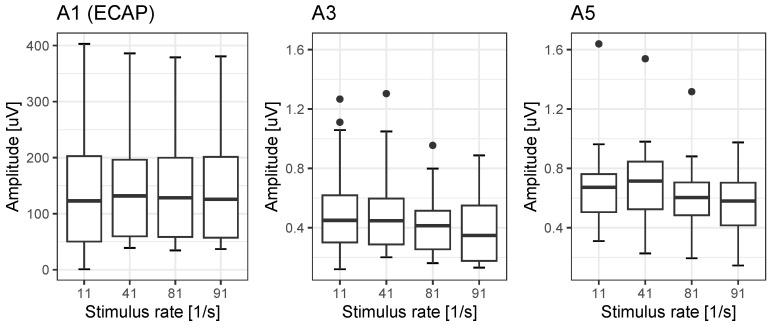
Amplitudes of all measured potentials. The boxes show medians and quartiles; the whiskers show the maximum value within 1.5 IQR. Filled circles show the outliers.

**Table 1 jcm-12-07188-t001:** Biographical data of participants.

ID	Age (Years)	Usage of CI (Months)	Side	Gender	Implant Type	Electrode Type	WRS_65_(CI) (%)
#098	62	146	r	f	CI512	CA	75.0
#107	62	140	l	f	CI512	CA	85.0
#140	59	104	r	f	CI24RE	CA	92.5
#160	65	97	r	f	CI24RE	CA	67.5
#193	58	81	r	f	CI422	SS	75.0
#234	57	71	l	m	CI522	SS	55.0
#242	53	26	l	f	CI522	SS	87.5
#247	53	25	r	m	CI532	SM	85.0
#251	72	52	l	f	CI532	SM	75.0
#262	45	61	l	f	CI512	CA	75.0
#269	55	13	l	f	CI512	CA	77.5
#271	73	8	l	m	CI532	SM	80.0
#275	73	3	r	m	CI532	SM	60.0
#279	62	28	r	f	CI532	SM	82.5
#281	56	27	r	f	CI532	SM	87.5
#288	57	19	r	f	CI622	SS	82.5
#315	38	7	r	f	CI622	SS	75.0
#341	64	3	r	m	CI622	SS	87.5
#348	74	116	l	f	CI612	CA	82.5
#350	59	1	l	f	CI622	SS	60.0

`Age’ is the recipient’s age at the time of inclusion in the study. Side is coded right (r) or center (l) for ear receiving the implant. Recipient’s sex is indicated, (f) or (m). Electrode types were: Contour Advance^®^ (CA), Slim Modiolar (SM), and Slim Straight (SS).

**Table 2 jcm-12-07188-t002:** Friedman rank sum test of latency t3 ($p=2\times 10^{-10}$), post hoc analysis.

	obs diff	critical ciff	stat signif	*p*
11–41	13.5	21.5	FALSE	0.98
11–81	30.5	21.5	TRUE	0.001
11–91	52.0	21.5	TRUE	$2\times 10^{-9}$
41–81	17.0	21.5	FALSE	0.37
41–91	38.5	21.5	TRUE	2×10−5
81–91	21.5	21.5	FALSE	0.08

**Table 3 jcm-12-07188-t003:** Friedman rank sum test of latency t5 ($p=3\times 10^{-10}$), post hoc analysis.

	obs diff	critical diff	stat signif	*p*
11–41	4.0	21.5	FALSE	1.0
11–81	29.5	21.5	TRUE	0.003
11–91	48.5	21.5	TRUE	3×10−8
41–81	25.5	21.5	TRUE	0.02
41–91	44.5	21.5	TRUE	$5\times 10^{-7}$
81–91	19.0	21.5	FALSE	0.19

**Table 4 jcm-12-07188-t004:** Friedman rank sum test of t5 − t1 interpeak interval ($p=3\times 10^{-10}$), post hoc analysis.

	obs diff	critical diff	stat signif	*p*
11–41	4.0	21.5	FALSE	1.0
11–81	29.5	21.5	TRUE	0.003
11–91	48.5	21.5	TRUE	3×10−8
41–81	25.5	21.5	TRUE	0.02
41–91	44.5	21.5	TRUE	$5\times 10^{-7}$
81–91	19.0	21.5	FALSE	0.20

**Table 5 jcm-12-07188-t005:** Friedman rank sum test of t5 − t3 interpeak interval ($p=1\times 10^{-8}$), post hoc analysis.

	obs diff	critical diff	stat signif	*p*
11–41	3.5	21.5	FALSE	1.0
11–81	29.5	21.5	TRUE	0.003
11–91	38.0	21.5	TRUE	$3\times 10^{-5}$
41–81	33.0	21.5	TRUE	$5\times 10^{-4}$
41–91	41.5	21.5	TRUE	$4\times 10^{-6}$
81–91	8.5	21.5	FALSE	1.0

**Table 6 jcm-12-07188-t006:** Friedman rank sum test of t3 − t1 interpeak interval ($p=2\times 10^{-10}$), post hoc analysis.

	obs diff	critical diff	stat signif	*p*
11–41	13.5	21.5	FALSE	0.99
11–81	30.5	21.5	TRUE	0.002
11–91	52.0	21.5	TRUE	$2\times 10^{-9}$
41–81	17.0	21.5	FALSE	0.37
41–91	38.5	21.5	TRUE	2×10−5
81–91	21.5	21.5	FALSE	0.08

**Table 7 jcm-12-07188-t007:** Friedman rank sum test of amplitude A3 ($p=9\times 10^{-5}$), post hoc analysis.

	obs diff	critical diff	stat signif	*p*
11–41	2.0	21.5	FALSE	1.0
11–81	21.0	21.5	FALSE	0.10
11–91	29.0	21.5	TRUE	0.004
41–81	23.0	21.5	TRUE	0.048
41–91	31.0	21.5	TRUE	0.001
81–91	8.0	21.5	FALSE	1.0

**Table 8 jcm-12-07188-t008:** Friedman rank sum test of amplitude A5 ($p=2\times 10^{-5}$), post hoc analysis.

	obs diff	critical diff	stat signif	*p*
11–41	16.0	21.5	FALSE	0.50
11–81	17.0	21.5	FALSE	0.37
11–91	19.0	21.5	FALSE	0.20
41–81	33.0	21.5	TRUE	$5\times 10^{-4}$
41–91	35.0	21.5	TRUE	$2\times 10^{-4}$
81–91	2.0	21.5	FALSE	1.0

## Data Availability

The data presented in this study are available on request from the corresponding author. The data are not publicly available due to data protection regulations.

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
