# Peer review of "The Effects of Stimulus Repetition Rate on Electrically Evoked Auditory Brainstem Potentials in Postlingually Deafened Adult Cochlear Implant Recipients"

_jcm, 2023, doi:10.3390/jcm12227188_

Round 1
Reviewer 1 Report
Comments and Suggestions for Authors
Dear Authors, I would like to thank you for your interesting article, anyway I've found some issues I would like to draw attention to.
1 You wrote that: To measure rate dependences of latencies and inter-peak latencies of EABRs, a quasi- simultaneous measurement of electrically evoked compound action potentials (ECAP)s and EABRs is needed" (line 100) As I understood you measure latencies between positive (?) peak of ECAP (t1) and t3 and t5 of EABR, but it is my suspicion - you could clarify it in the article.
2 Did you use the same pulse duration (100us) for ECAP and EABR? And the same question about stimulation electrode for ECAP - 11 vs 18 like EABR? If I am correct please write it in plain text, because it is not default settinng.
On the other hand you wrote: "ECAP was measured by stimulating all intra cochlear electrodes"... So if you stimulate all electrodes, why not 11th and 18th?
3 You mentioned: "This approach led to improved differential diagnosis for CI recipients 58 and improved intraoperative assessment" (line 58). What kind of introp measurement?
4 You show outliers in Fig 2 and Fig 3. So if there are outliers in patients with good results, I wonder if EABR results for poor performers would be better or worse than these outliers? Or if your method of finding EABR anomalies would be accurate and specific enough. Did you measured any patients from poor performers group?
Author Response
Reviewer #1
Comments and Suggestions for Authors
Dear Authors, I would like to thank you for your interesting article, anyway I've found some issues I would like to draw attention to.
- You wrote that: To measure rate dependences of latencies and inter-peak latencies of EABRs, a quasi- simultaneous measurement of electrically evoked compound action potentials (ECAP)s and EABRs is needed" (line 100) As I understood you measure latencies between positive (?) peak of ECAP (t1) and t3 and t5 of EABR, but it is my suspicion - you could clarify it in the article.
Thank you very much for the detection of vagueness. We added the following sentence at the end of this paragraph to clarify this.
[…] Since ECAP and EABR are recorded with opposite polarities, the inter-peak latencies were determined between the negative peak of the ECAP and the corresponding positive peaks in the EABR, measured at the same stimulation intensity. […]
- Did you use the same pulse duration (100us) for ECAP and EABR? And the same question about stimulation electrode for ECAP - 11 vs 18 like EABR? If I am correct please write it in plain text, because it is not default settinng.
On the other hand you wrote: "ECAP was measured by stimulating all intra cochlear electrodes"... So if you stimulate all electrodes, why not 11th and 18th?
Yes, you are right. We used the same stimulation mode for ECAP and EABR still using electrode 11 as stimulation-active and electrode 18 as stimulation indifferent electrode. In order to remove contradictions, we corrected the first sentence in section 2.2.2.
[…] All EABR measurement series were performed in the same stimulation mode as for the rate-dependent ECAP using the EABRCIStim described above. […]
- You mentioned: "This approach led to improved differential diagnosis for CI recipients and improved intraoperative assessment" (line 58). What kind of introp measurement?
Thank you for your comment. We have clarified the sentence with an addition.
[…] This approach led to improved differential diagnosis for CI recipients and improved intraoperative assessment by using objective methods like electrically evoked compound action potentials (ECAP) and EABR […]
- You show outliers in Fig 2 and Fig 3. So if there are outliers in patients with good results, I wonder if EABR results for poor performers would be better or worse than these outliers? Or if your method of finding EABR anomalies would be accurate and specific enough. Did you measured any patients from poor performers group?
Thank you for your comment. All 20 ears met the defined inclusion criteria at the time of measurement and were therefore not poor performers. It is possible that this is a display problem. As the whiskers of our boxplots show the largest value within 1.5 times the standard deviation according to John W. Tukey's definition, values further outside the standard deviation are shown as outliers. According to this definition, outlying values can be treated as suspected outliers. If the 5% percentiles were displayed, most of the outliers would probably lie within this range.
Measuring poor performing patients and comparing them to a good performing cohort is an interesting future project.
Reviewer 2 Report
Comments and Suggestions for Authors
The current manuscript provided the values of electrically evoked brainstem responses (EABRs) in terms of rate-dependent latencies and amplitudes in 18 postlingually deaf patients (20 ears) who underwent cochlear implantations. The manuscript is well-written, but there are some issues that need to be addressed:
- Please provide more data concerning sample size calculation
- In the Abstract and Section 2.1 it was mentioned that 20 patients are included in the study, but Table 1 summarized the information of 18 patients (20 ears)
- The description of abbrevations in Table 1 should come at the bottom of the table
- The names of the devices used in electrophysiological tests should be specified in materials and methods
- Please add the summary of the features of the study population (such as mean age and SD)
- Some abbreviations are not explained in the text, such as t1 and A1
- What post-hoc test was used?
- The first columns in Tables 2-6 need definition. Abbreviations used in the first rows of Tables 2-6 should be explained at the bottom of the tables
Author Response
Reviewer #2
Comments and Suggestions for Authors
The current manuscript provided the values of electrically evoked brainstem responses (EABRs) in terms of rate-dependent latencies and amplitudes in 18 postlingually deaf patients (20 ears) who underwent cochlear implantations. The manuscript is well-written, but there are some issues that need to be addressed:
- Please provide more data concerning sample size calculation
Thank you for addressing this. We added the data of our power calculation.
- In the Abstract and Section 2.1 it was mentioned that 20 patients are included in the study, but Table 1 summarized the information of 18 patients (20 ears)
Thank you for your observation. We corrected this mistake.
- The description of abbrevations in Table 1 should come at the bottom of the table
We appreciate your observation on the layout and we have adapted it accordingly. We changed the layout of table 1.
- The names of the devices used in electrophysiological tests should be specified in materials and methods
We originally wrote: “The Eclipse system (Interacoustics, Middelfart, Denmark) was used to record the rate-dependent EABRs.”
We assume that this sufficiently describes the measuring device.
- Please add the summary of the features of the study population (such as mean age and SD)
We added the population summary.
- Some abbreviations are not explained in the text, such as t1 and A1
In section 3.1. we opened with "The latencies t1, t3 and t5 of rate-dependent..." and similarly in sections 3.2. and 3.3. The explanations of the abbreviations are, in our opinion, sufficiently given. In addition, we added a sentence in the text for a more detailed description.
[…] The labelling and numbering of the marked potentials was performed according to Jewett and Williston […]
- What post-hoc test was used?
Good point. As a post-hoc test, we used the test of multiple comparison after Friedman test. We added this sentence to section 2.3.
- The first columns in Tables 2-6 need definition. Abbreviations used in the first rows of Tables 2-6 should be explained at the bottom of the tables
Thank you for pointing this out. We added the abbreviations into the List of abbreviations at the end of the manuscript.

Round 2
Reviewer 2 Report
Comments and Suggestions for Authors
I would like to thank the authors for addressing all the issues in the manuscript.
Author Response
Reviewer #2
Comments and Suggestions for Authors
I would like to thank the authors for addressing all the issues in the manuscript.
Thank you very much. We are pleased that the quality of our manuscript has improved significantly thanks to your tips.
